# Portuguese Validation of a Reduced Version of the IAT (Internet Addiction Test) Scale—Youth Version

Ivone Patrão [1,2,3], Inês Borges [2,3,4], Patrícia Sobral [2,3] and Ana Moreira [1,2,5,*]

1 APPsyCI—Applied Psychology Research Center Capabilities & Inclusion, ISPA—Instituto Universitário, 1149-041 Lisboa, Portugal; ivonepatrao@ispa.pt
2 School of Psychology, ISPA—Instituto Universitário, Rua do Jardim do Tabaco 34, 1149-041 Lisboa, Portugal; ines.reis.borges@gmail.com (I.B.); patriciasobral.psi@gmail.com (P.S.)
3 Geração Cordão, 1990-601 Lisboa, Portugal
4 WJCR—William James Center for Research, ISPA—Instituto Universitário, 1149-041 Lisboa, Portugal
5 Faculdade de Ciências e Tecnologia, Universidade Europeia, 1500-210 Lisboa, Portugal
* Correspondence: amoreira@ispa.pt

**Abstract:** This study adapted and assessed a shortened version of the IAT (Internet Addiction Test) scale completed by young people aged 12 years and older regarding their online behaviors and risk of online addiction. The psychometric qualities of the reduced version (Screening IAT—youth) are presented in order to validate the use of this version in the early detection of online addiction. The total sample was composed of 3021 participants that were 55.9% female and 44.1% male, with a mean age of $\bar{x}$ = 15 years ($\sigma$ = 3.028), attending junior high school (56.2%), high school (37.8%), and college (5.9%). The procedure included a factorial analysis in which the total sample was randomly divided into three samples. An exploratory factor analysis was performed with one part of the sample, and a confirmatory factor analysis was performed with the other parts, assessing internal consistency, construct reliability, and discriminant validity. The results indicate that this reduced version of the IAT for young people has good psychometric qualities and that it can be applied in research and clinical settings. With this version and the parent–teacher version, there is a direct contribution to a tripartite assessment of internet addiction risk.

**Keywords:** internet addiction test; reduced version; factorial analysis; youth





## 1. Introduction

We are currently living in a new digital age in which the internet is the most accessible and utilized means of communication, especially for the younger generations, where they can enjoy greater ease of access through digital platforms that contain chats, games, photo posting, etc. [1]. Kimberly Young and Mark Griffiths were the first authors to explore the impact that new technologies would have on society, eventually developing the concept of internet addiction [2,3].

According to [4], internet addiction encompasses a variety of online behaviors that are considered pathological when individuals are not able to regulate them, such as a compulsion to shop or play online games (in which money is invested), excessive surfing of web pages, online video game addiction, and cybersex/pornography. Online addiction, considered by many authors to be problematic internet use, has severe implications for both physical health (e.g., body posture problems and tendonitis) and mental health (e.g., depression and social isolation) and the development of young people [5,6]. Concerning internet addiction according to gender, ref. [7] states that girls are more likely to have higher levels of online addiction. However, ref. [8] establishes some differences between girls and boys, stating that boys have a higher dependence on online games and girls have a higher dependence on social networks. According to [9], the prevalence of online addiction in general is higher in boys, but when it comes to social networks, this prevalence is higher

in girls. As for age, ref. [10] states that young people between 15 and 16 are most dependent on the internet.

Having said that, ref. [4] developed an instrument entitled the Internet Addiction Test (IAT) consisting of a set of questions such as "Do you feel worried about the Internet?"; "Do you stay online longer than you initially intended?"; and "Do you feel restless, depressed or irritable when you try to cut down or stop using the Internet?", among others, with the purpose of assessing internet addiction. According to the criteria in [4], subjects answering affirmatively to five or more of these questions would be considered dependent users. For [2,3], internet addiction involves not only spending many hours online but also compromising the normal functioning of the subject in society, namely through uncontrolled and unconscious interactions. However, it should be noted that moderate and controlled internet use does not present significant risks for individuals since it allows for remote socialization and the creation or strengthening of new relationships [2,3,11].

Research on online addiction has grown internationally and nationally in recent years. Previous studies have issued warnings about the incidence of cases of young people at risk of online addiction, which implies the need for continuous and deeper study of the explanatory variables of this phenomenon [12].

A study conducted in a partnership between the APAV (Portuguese Association for Victim Support) and Geração Cordão to assess online risk behaviors and the impact of internet use on mental health in a sample of young Portuguese through the data presented in descriptive and inferential statistics indicated which characteristics are associated with a risk profile of technology and internet use in this sample. The most prevalent characteristics of young people who present an online or smartphone addiction are presented in Table 1 [13].

**Table 1.** The most prevalent characteristics in young people with internet addictions.

| | |
|---|---|
| 1 | Young people between 16 and 21 years old, males, and those attending high school |
| 2 | No physical exercise |
| 3 | School performance—students with negative grades |
| 4 | No loving relationship |
| 5 | Began online activity at age 8 |
| 6 | Use screens at night (fall asleep with technology and sleep with technology at night) |
| 7 | More than six hours per day online |
| 8 | Make meals/snacks when online |
| 9 | Take time away from other offline activities to spend more time online (being on the internet takes time away from being with family, socializing, studying, sleeping, dating, exercising, and playful activities) |
| 10 | Send and receive nudes |
| 11 | Practice sexting |
| 12 | Play online games |
| 13 | Present changes in sleep, mood, and cyberbullying—victim and/or aggressor |
| 14 | Have more episodes of online discomfort |
| 15 | Do not talk to anyone when faced with situations of online discomfort |
| 16 | Phubbing—victim or phubber |
| 17 | Ghosting—victim and ghost |
| 18 | Low empathy |
| 19 | Perception of parenting style—permissive |

Some of these typical characteristics of young people with online addictions (Table 1) are common to the findings of the study by [14].

In addition to the Internet Addiction Test (IAT) developed by [15], several instruments allow us to study internet addiction (Table 2).

**Table 2.** Examples of instruments to assess internet addiction.

| Instrument | Number of Items | Authors |
|---|---|---|
| The Internet Addiction Questionnaire (IAQ) | 30 | [16] |
| The Internet Addiction Scale (IAS) | 31 | [17] |
| The Problematic Internet Use Questionnaire (PIUQ) | 58 | [18] |
| The Adolescent Pathological Internet Use Scale (APIUS) | 11 | [19] |
| The Compulsive Internet Use Scale (CIUS) | 178 | [20] |
| EUPI-a: Escala de Uso Problemático de Internet en adolescentes | 11 | [21] |

However, in a literature review by [22], the most used instrument in studies on this subject is the Internet Addiction Test, with 1096 citations. It is also the most used instrument in different countries. Consulting Scopus reveals that the article containing this instrument has 2911 citations.

Some of the countries who have translated, adapted, or used this instrument are listed in Table 3.

**Table 3.** Studies carried out in different countries using the IAT scale.

| Country | Authors | Target Population |
|---|---|---|
| Portugal | [23] | Individuals 15–39 years old |
| France | [24] | Adults |
| Germany | [25] | University students |
| Spain | [26] | University students |
| Italy | [27] | Individuals over 18 years old |
| Greece | [28] | Teenage students |
| Turkey | [29] | University students |
| Croatia | [30] | Students (15–20 years) |
| China | [31] | High school students |
| South Korea | [32] | Students aged 15 and over |
| Thailand | [33] | University students (medicine) |
| Indonesia | [34] | Teenagers |
| Japan | [35] | University students |
| United States | [36] | Young adults (21–28 years) |
| Lebanon | [37] | University medical students |
| China | | School children |
| Spain | [38] | Women with eating disorders |
| Malaysia | | University students |

Therefore, the existence of an instrument to screen for internet addiction has become increasingly relevant in clinical practice so as to prevent online risk behaviors and the impact of internet use on mental health.

Having said this, the aim of this study was to adapt and evaluate a shortened version of the IAT (Internet Addiction Test) scale completed by young people aged 12 years and older in relation to their online behaviors and risk of online addiction. The psychometric qualities of the reduced version (Screening IAT—young people) are presented in order to validate the use of this version in the early detection of online addiction. The following research hypothesis was formulated:

**Hypothesis 1.** *The reduced version of the IAT scale has good psychometric qualities.*

## 2. Methods

### 2.1. Procedure

This study involved the voluntary participation of 3021 individuals, and the questionnaire was completed by young people between 12 and 25 years old. This study's procedures were carried out according to the Helsinki Declaration. The ISPA (Instituto Universitário

Ethics Committee (I/001/03/2018)) and the Portuguese Ministry of Education approved this study. Concerning students under 18, their parents signed an informed consent form, giving them permission to participate in this study. It should be noted that contact with these participants was made through their school, which was included in the Geração Cordão program. The participants over 18 years old signed the informed consent form. Therefore, the sampling process was the non-probabilistic, convenient, and intentional snowball type [39] (Trochim 2000).

The online questionnaire contained information about the objective of this study, as well as the confidentiality of the answers. The online questionnaire was composed of sociodemographic questions in order to characterize the sample and the instrument we propose to validate for the Portuguese population. Data collection took place from 2021 to 2022 in five school groups in Continental Portugal and the Autonomous Region of Madeira.

### 2.2. Participants

The sample in this study consisted of 3021 participants between the ages of 12 and 25, with an average age of 15 (SD = 3.03). Of these participants, 55.9% were female and 44.1% were male. Their educational qualifications included junior high school (56.2%), high school (37.8%), and college (5.9%).

### 2.3. Data Analysis Procedure

Once the data were collected and the database was created in SPSS Statistics 29 software (IBM Corp., Armonk, NY, USA), the metric qualities of the sample used in this study were tested. The first step was to divide the sample into three parts (20%, 40%, and 40%). This decision was made because when an instrument is to be adapted or reduced, the sample must be large enough to be divided into three parts [40]: with 20% of the sample, an exploratory factor analysis must be carried out; with 40%, a confirmatory factor analysis; and with the remaining 40%, another confirmatory factor analysis. The exploratory factor analysis examined related variables to create a measurement scale for controlling factors influencing the original variables [40]. The other two parts (1200 and 1221 participants) were then tested for validity by performing a confirmatory factor analysis (CFA) for each of the groups using AMOS 29 for Windows software (IBM Corp., Armonk, NY, USA). The procedure followed the logic of "model generation" [41], considering in the analysis of their fit, in an interactive way, the results obtained: for the chi-square ($\chi^2$) $\leq$ 5; for the Tucker–Lewis index (TLI) > 0.90; for the goodness-of-fit index (GFI) > 0.90; for the comparative fit index (CFI) > 0.90; for the root-mean-square error approximation (RMSEA) $\leq$ 0.08; and for the root-mean-square residual (RMSR), a smaller value corresponded to a better adjustment [42]. Obtaining a good fit for both measurement models was essential to establish the convergent validity of each model and to verify the risks associated with common variance methods [43]. However, when the Cronbach's alpha value is above 0.70, AVE values greater than 0.40 are acceptable, indicating good convergent validity [44]. An invariance analysis between two groups randomly selected from the whole sample was also performed. Then, its internal consistency was tested by calculating Cronbach's alpha, and finally, a sensitivity study was carried out by calculating measures of central tendency such as the median, asymmetry, and kurtosis, as well as the minimum and maximum for each item.

The effects of sociodemographic variables on internet addiction were also tested. A Student's t-test for independent samples was used for gender, and a one-way parametric ANOVA was used for schooling. The association between age and internet addiction was studied using Pearson's correlations. We then created internet addiction scores and tested whether gender and schooling were independent of internet addiction.

### 2.4. Instrument

The instrument used in this study was a reduced version of the Internet Addiction Test (IAT) developed by [45] and adapted to the Portuguese population by [23]. The

initial instrument was composed of 20 items measured on a 6-point Likert scale 'does not apply' (0), 'rarely' (1), 'occasionally' (2), 'frequently' (3), 'often' (4), and 'always' (5). Only seven items were used in this study, considering the items that presented higher factor weights in both the Portuguese version and subsequent studies. The following items were used: 1, 8, 13, 15, 18, 19, and 20. In this study, they were numbered 1 to 7 (Table 4).

**Table 4.** Items of the short version of the instrument.

| Numbering in the 20-Item Scale | Numbering in the 7-Item Scale | How Often... |
|:---:|:---:|:---:|
| 1 | 1 | Do you find that you stay on-line longer than you intended? |
| 8 | 2 | Does your performance or productivity at school/study/work suffer because of the Internet? |
| 13 | 3 | Do you snap, yell, or act annoyed if someone bothers you while you are on-line? |
| 15 | 4 | Do you feel preoccupied with the Internet when off-line, or fantasize about being on-line? |
| 18 | 5 | Do you try to hide how long you've been on-line? |
| 19 | 6 | Do you choose to spend more time on-line over going out with others? |
| 20 | 7 | Do you feel depressed, moody, or nervous when you are off-line, which goes away once you are back on-line? |

## 3. Results

### 3.1. Validity, Reliability, and Sensitivity of the Instrument

To perform the factorial analysis, the total sample was randomly divided into three samples. In the first sample, 600 participants were extracted; in the second, 1200 participants were extracted; and in the third, 1221 participants were extracted.

An exploratory factor analysis was carried out using the initial sample. According to the first exploratory factor analysis, the scale was formed by a single factor (unidimensional) with a KMO of 0.86, which was good [46] (Sharma, 1996), and the results of Bartlett's test of sphericity were significant at $p < 0.001$, indicating that the data were from a multivariate normal population [47] (Pestana & Gageiro, 2003). The factor structure of this scale was based on a factor that explained 51% of the total variability of the scale. All items had factor weights above 0.50 (Table 5). As for internal consistency, Cronbach's alpha was 0.83.

**Table 5.** Item loadings.

| | How Often... | Loadings |
|:---:|:---:|:---:|
| 1 | Do you find that you stay on-line longer than you intended? | 0.55 |
| 2 | Does your performance or productivity at school/study/work suffer because of the Internet? | 0.68 |
| 3 | Do you snap, yell, or act annoyed if someone bothers you while you are on-line? | 0.73 |
| 4 | Do you feel preoccupied with the Internet when off-line, or fantasize about being on-line? | 0.76 |
| 5 | Do you try to hide how long you've been on-line? | 0.71 |
| 6 | Do you choose to spend more time on-line over going out with others? | 0.71 |
| 7 | Do you feel depressed, moody, or nervous when you are off-line, which goes away once you are back on-line? | 0.83 |

In the subsequent confirmatory factor analysis carried out with a sample of 1200 participants, the obtained adjustment indices proved to be adequate ($\chi^2/gl$ = 2.93; GFI = 0.99; CFI = 0.99; TLI = 0.99; RMSEA = 0.040; SRMR = 0.029). The internal consistency presented a Cronbach's alpha of 0.84. It also showed good construct reliability with a value of 0.85 and convergent validity with a value of 0.45.

Figure 1 shows the factorial weight as well as the individual reliability of each of the items of the sample with 1200 participants.

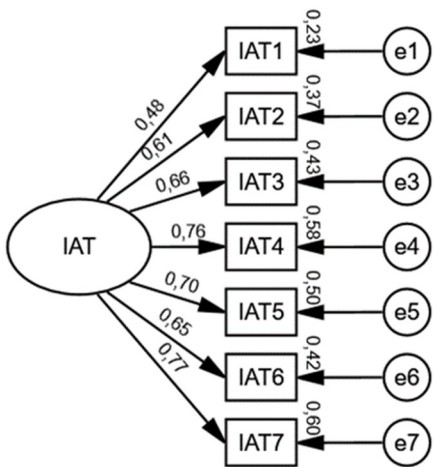

**Figure 1.** Factorial weight and individual reliability of each of the items of the sample with 1200 participants.

Concomitantly, in the confirmatory factor analysis carried out with a sample of 1221 participants, the obtained adjustment indices were adequate ($\chi^2/gl = 2.97$; GFI = 0.99; CFI = 0.99; TLI = 0.98; RMSEA = 0.040; SRMR = 0.025). It also showed good construct reliability with a value of 0.82 and convergent validity with a value of 0.40.

Figure 2 presents the factorial weight and the individual reliability of each item of the sample with 1221 participants.

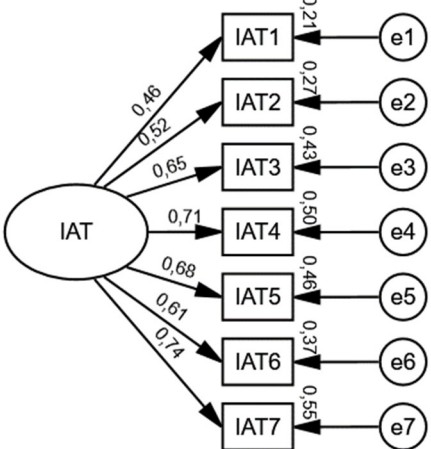

**Figure 2.** Factorial weight and individual reliability of each of the items of the sample with 1221 participants.

Next, the invariance of the Internet Addiction Test model in boys and girls was tested and assessed by comparing the free model (with factor weights and variances/covariances of the free factors) with the construct model in which the factor weights and variances/ covariances of both groups were fixed. The significance of both models was measured using the chi-square test described by [48]. The restricted model, with factorial weights and variances/covariances fixed between the two groups, did not show a significantly worse fit than the model with free parameters ($\Delta\chi 2\lambda(6) = 18.12$; $p = 0.229$), which indicated the invariance of the measurement model of the Internet Addiction Test between boys and girls. We also found that the intercepts were invariant ($\Delta\chi 2i(7) = 10.768$; $p = 0.149$), which indicated that we were looking at a model with strong invariance.

The scale's internal consistency was tested among the 3021 participants, obtaining a Cronbach's alpha of 0.83.

Concerning item sensitivity, it was found that none of the items had a median close to one of the extremes, all items had responses at all points, and their absolute values of skewness and kurtosis were below two and seven, respectively (Table 6), which indicated that they did not grossly violate normality [49]. As far as the scale was concerned, it did not grossly violate normality since its absolute skewness and kurtosis values were below two and seven, respectively [49].

**Table 6.** Measurement of central tendency, skewness, and kurtosis of the items.

|  | Median | Skewness | Std. Error of Skewness | Kurtosis | Std. Error of Kurtosis | Minimum | Maximum |
|---|---|---|---|---|---|---|---|
| Do you find that you stay on-line longer than you intended? | 3.00 | −0.282 | 0.045 | −0.548 | 0.089 | 0 | 5 |
| Does your performance or productivity at school/study/work suffer because of the Internet? | 2.00 | 0.652 | 0.045 | 0.062 | 0.089 | 0 | 5 |
| Do you snap, yell, or act annoyed if someone bothers you while you are on-line? | 1.00 | 1.106 | 0.045 | 1.067 | 0.089 | 0 | 5 |
| Do you feel preoccupied with the Internet when off-line, or fantasize about being on-line? | 1.00 | 1.265 | 0.045 | 1.983 | 0.089 | 0 | 5 |
| Do you try to hide how long you've been on-line? | 1.00 | 1.090 | 0.045 | 0.990 | 0.089 | 0 | 5 |
| Do you choose to spend more time on-line over going out with others? | 1.00 | 1.034 | 0.045 | 0.765 | 0.089 | 0 | 5 |
| Do you feel depressed, moody, or nervous when you are off-line, which goes away once you are back on-line? | 1.00 | 1.249 | 0.045 | 1.679 | 0.089 | 0 | 5 |

### 3.2. Prevalence of Internet Addiction in the Sample

According to [15], the IAT instrument, composed of 20 items, distinguished three types of users according to their different levels of internet use. To this end, they created three cut-off points: 20–39 = average user; 40–69 = a person who has frequent problems due to their internet use; and 70–100 = internet addicts. In [23], the authors used the same cut-off points in their adaptation to the Portuguese population. However, as the reduced version proposed in this study is composed of only seven items, the proposed cut-off points are 0–11 = low risk of addiction; 12–19 = moderate risk of addiction; and 20–35 = high risk of addiction.

According to the proposed cut-off criteria, 1710 (56.6%) of the participants in this study showed a low risk of internet addiction, 1051 (34.8%) showed a moderate risk of internet addiction, and 260 (8.6%) showed a high risk of internet addiction.

### 3.3. Effects of Sociodemographic Variables on Internet Addiction

Finally, the effects of sociodemographic variables on internet addiction were tested. A Student's t-test for independent samples was used for gender. Female participants were found to have higher levels of internet addiction than male participants (Table 7). However, these differences were not statistically significant (t (3019) = 0.90; $p = 0.364$; d = 0.03).

**Table 7.** Effects of sociodemographic variables on internet addiction.

| Variable | | Mean | SD |
|---|---|---|---|
| Gender | Female | 11.34 | 5.70 |
|  | Male | 11.14 | 6.00 |
| Education | Junior high school | 11.32 | 6.28 |
|  | High School | 11.20 | 5.24 |
|  | College | 10.88 | 4.97 |

A one-way parametric ANOVA was used for the level of education. The participants with junior high school education revealed higher levels of internet addiction, followed by those with high school education and, finally, those attending college (Table 7). Nevertheless, these differences were not statistically significant (F (3, 3018) = 0.51; $p = 0.600$).

The association between age and internet addiction was also tested using Pearson's correlations.

The results show that age was negatively and significantly associated with internet addiction (r = −0.036; $p$ = 0.046), which means that younger participants showed greater internet addiction (Table 8).

**Table 8.** Correlation between age and internet addiction.

| | | Age | Internet Addiction |
|---|---|---|---|
| Age | Pearson Correlation | -- | |
| | N | 3021 | |
| Internet Addiction | Pearson Correlation | −0.036 * | -- |
| | Sig. (2-tailed) | 0.046 | |
| | N | 3021 | 3021 |

Note: * $p$ < 0.05.

We then tested whether sociodemographic variables were independent of internet addiction. To this end, the chi-squared test of independence was used.

The percentage of female participants with a low risk of internet addiction was similar to the percentage for males. This was also true for a moderate risk of internet addiction. However, the percentage of participants with a high risk of internet addiction was higher in female participants when compared with male participants (Table 9). The chi-squared test ($\chi^2$ (2) = 1.18; $p$ = 0.555; V = 0.02) suggested that the two variables were independent.

**Table 9.** Gender × IAT crosstab.

| | | Gender | | Total |
|---|---|---|---|---|
| | | Female | Male | |
| IAT | Low risk of addiction | 961 | 749 | 1710 |
| | | 56.9% | 56.3% | 56.6% |
| | Moderate risk of addiction | 577 | 474 | 1051 |
| | | 34.1% | 35.6% | 34.8% |
| | High risk of addiction | 152 | 108 | 260 |
| | | 9.0% | 8.1% | 8.6% |
| | Total | 1690 | 1331 | 3021 |
| | | 100.0% | 100.0% | 100.0% |

When testing the independence of the risk of internet addiction and the level of education, it was found that these variables were not independent ($\chi^2$ (4) = 16.07; $p$ = 0.003; V = 0.07). The youngsters attending university had a higher percentage with a low risk of internet addiction. However, they had the lowest percentage with a moderate or high risk of internet addiction. The participants in secondary school had a higher percentage with a moderate risk of internet addiction. As for the high risk of internet addiction, the participants in primary schools were revealed to have a higher percentage (Table 10).

**Table 10.** Education × IAT crosstab.

| | | Education | | | Total |
|---|---|---|---|---|---|
| | | Junior High School | High School | College | |
| IAT | Low risk of addiction | 945 55.6% | 651 57.0% | 114 63.7% | 1710 56.6% |
| | Moderate risk of addiction | 580 34.1% | 416 36.4% | 55 30.7% | 1051 34.8% |
| | High risk of addiction | 174 10.2% | 76 6.6% | 10 5.6% | 260 8.6% |
| | Total | 1699 100.0% | 1143 100.0% | 179 100.0% | 3021 100.0% |

## 4. Discussion

The main objective of this study was to validate a reduced version of the instrument developed by [45] and adapted for the Portuguese population by [23], consisting of 20 items. This instrument measures internet addiction. The version proposed in this article is a reduced version composed of seven items, which will be applied to young people attending primary schools, secondary schools, and higher education. We chose the items that are consistent with the general criteria of internet addiction (e.g., tolerance, mood change, and interpersonal conflict) and presented higher factor weights in both the version adapted to the Portuguese population and subsequent studies. This selection was reinforced by an assessment performed by one of the researchers of this study in her clinical practice, who detected that these items are the most relevant.

As previously mentioned, the sample of this study was composed of 3021 participants. For scale validation, the sample was divided into three parts. An exploratory factor analysis was performed with 600 participants and obtained a good KMO value [46]. More than 0.50 of the total variance was explained, and all items had factor weights greater than 0.50.

The other parts into which the sample was divided consisted of 1200 and 1221 participants. Two one-factor confirmatory factor analyses were carried out with these two samples. The obtained adjustment indices are all adequate [42]. Comparing the CFA results obtained in this study with those of the study carried out by [27], it can be seen that the adjustment indices obtained in this study are better. It should be noted that the sample size in [27] was much smaller than the sample in this study (n = 463), and the average age of the participants was also higher. On the other hand, the adjustment indices are identical to those of the study carried out by [38]. They show good construct reliability, although their convergent validity is slightly below 0.50. However, when the Cronbach's alpha value is above 0.70, AVE values greater than 0.40 are acceptable, indicating good convergent validity [44].

Finally, we randomly divided the whole sample into two parts to perform a gender invariance analysis, which confirmed that we were studying a model with strong invariance. These results align with those obtained by [38], who confirmed invariance according to the participant's gender.

Regarding the internal consistency, the scale has a Cronbach's alpha of 0.83, which can be considered good [50]. All items show good sensitivity, which indicates that they can discriminate subjects. The Cronbach's alpha value obtained in this study is higher than that of [27], indicating better internal consistency.

The effects of sociodemographic variables on internet addiction showed us that female participants had more internet addiction than male participants, which may have been related to the social network consumption profile associated with the female gender. This profile is more normalized, as it is a source of contact with others and socialization. These results align with the findings of some authors, such as [7], who found that girls are more

likely to experience online addiction than boys. However, boys may experience more problems with online games, while girls prefer social networks [8].

Concerning the level of education, the participants at primary schools had higher levels of internet addiction. This result is related to difficulties in the self-control of behavior and emotions during development. Throughout a school career and its different demands and challenges, an increasing learning curve of self-control of behavior and emotions can be expected.

Concerning age, there was a negative and significant correlation between age and internet addiction. These results are in line with a study carried out by [10], who tells us that they found the highest internet addiction among young people between the ages of 15 and 16 (our sample was composed of participants between the ages of 12 and 25).

The main limitations of this study are the fact that it used a self-report instrument and that we were dealing with participants over 12 years of age since in this area of internet addiction, from a clinical point of view, young people tend to perceive contact and the acceptable use of technology without assessing the impact on their daily functioning (e.g., impact on sleep routines, eating, and studying). It should be noted that young people at this age are often unaware of their dependence on the internet.

Another limitation of this study is the small number of sociodemographic questions included in the questionnaire. As in previous studies, among other questions, it would have been interesting to ask about the frequency and duration of time spent on the internet.

Another suggestion is to recommend that parents read the "Practical guide to the healthy use of technology" developed by the project of Geração Cordão [51]. This instrument, since it has few items, could be beneficial in the early diagnosis of internet addiction in young adolescents.

## 5. Conclusions

In conclusion, the results of the metric qualities of this instrument indicate that it may be used in future empirical studies and in clinical and educational settings to assess the risk of online addiction. Thus, a shorter instrument is available, which is very important for the population for which it is intended since we know that young people often need more time to answer very long instructions. For this young population, the instrument to which they must respond must be short.

In addition, in its shortened version, this tool is helpful as a test to quickly and effectively detect moderate-risk situations that allow for preventive intervention with a young person, their family, and the school context since by establishing a relationship of trust with the school and the family, problematic internet use behaviors can be prevented [6].

**Author Contributions:** Conceptualization, I.P., I.B. and P.S.; methodology, I.P., I.B. and A.M.; software, P.S. and A.M.; validation, I.P., I.B., P.S. and A.M.; formal analysis, P.S. and A.M.; investigation, I.P., I.B. and P.S.; resources, I.P., I.B. and P.S.; data curation, P.S. and A.M.; writing—original draft preparation, I.P., I.B. and P.S.; writing—review and editing, I.P., I.B. and P.S.; visualization, I.P.; supervision, I.P.; project administration, I.P.; funding acquisition, I.P. All authors have read and agreed to the published version of the manuscript.

**Funding:** This research received no external funding.

**Institutional Review Board Statement:** The study procedures were carried out according to the Helsinki Declaration. The ISPA (Instituto Universitário Ethics Committee) and the Portuguese Ministry of Education approved this study(protocol code I/001/03/2018 and approved on 2 April 2018). All participants were informed about this study, and they all signed an informed consent form. Regarding minors under 18 years of age, their parents signed the informed consent.

**Informed Consent Statement:** Informed consent was obtained from all subjects involved in this study.

**Data Availability Statement:** The data presented in this study are available on request from the corresponding authors. The data are not publicly available since, in their informed consent, participants were informed that the data were confidential and that individual responses would never be known, as data analysis would be performed with all participants combined.

**Conflicts of Interest:** The authors declare no conflicts of interest.

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
