# Peer review of "Portuguese Validation of a Reduced Version of the IAT (Internet Addiction Test) Scale—Youth Version"

_2673-5318, doi:10.3390/psychiatryint5010007_

Round 1

Reviewer 1 Report

Comments and Suggestions for Authors

In this study, the authors have examined the psychometric properties of the reduced version of the Internet Addiction Test (IAT) among Portuguese youth.

Overall, the manuscript needs to be improved significantly in content and writing quality (The manuscript could benefit from a revision by a native English speaker). More specifically, the manuscript lacks a thorough literature review on the IAT, which is needed for the Introduction and for discussing the results of the current study. Also, the authors need to clearly explain their research question, data analysis plan and present their results more clearly. The discussion also lacks depth. I have outlined my recommendations for the authors in the following:

Title:
The title could be revised to: ‘’
Portuguese validation of a reduced version of the IAT (Internet Addiction Test) scale - Youth version’’

Introduction:
- Overall, Intro is too short and it should include more critical information. For instance, there is no explanation on whether the IAT has been examined in other cultures and countries; what were the results? This is very important that authors provide a literature review of the IAT in the Introduction.
- The list of the most prevalent characteristics (Patrão et al., 2023) of young people who present online and smartphone addiction is not adequate in this form. It might be better in a table, but it would be more useful to go in-depth into the psychometric results: e.g. referring to articles such as Moon et al., 2018.

-What did the original study yield on the psychometrics of the measure? Did consequent studies replicate these results? Or was there any need for modification/item removal? All these are needed to be included in the Introduction.
-Have previous studies shown cross-cultural differences in IAT construct?
- Please also provide some hypotheses on what you expect about the factor structure and youth version of IAT and statistics that you have used to examine the validity of the IAT.

Method and Results:
- The item selection based on the highest factor loadings is not necessarily the best solution, as the youngest in that sample was 15 years old, and here it is 12. The reliability of self-reporting at age 12 is questionable, so the validity of the IAT should be much better supported, because the 7-item factor structure is not enough to confirm the validity of the IAT. It would have been worthwhile even to compare the measurement invariance of the 12-14 years old group with that of the older ones. I think this is more important than gender differences.     

-The demographic variables that are compared across groups are interesting. But please first provide information about these variables in the introduction and explain how they are related to the IAT construct. It would be strange for a reader to see these analyses while there are no explanations in the Introduction.
- The prevalence of Internet addiction in the sample is not evaluated, as the diagnostic reliability of the 7-item IAT scale is unknown. In the further results section, it would be more appropriate to write about the validity of this measure among early adolescents (12-14 year olds). It could have been supported by qualitative research (e.g. cognitive interview). 

- Since normality is violated for several items, it is also important to describe the method of factor analysis (eg. MLR).

Discussion:

- Discussion lacks depth and is too weak. The findings should be interpreted in the context of the results from previous studies. In the current format, the Discussion is mostly a repetition of the results section. What results did previous studies on the psychometric of the IAT yield? What about convergent validity? A literature review on the IAT is needed in the Introduction which could be used to discuss the results from the current study in the Discussion part.
- Please discuss your findings in the context of previous findings while also trying to avoid only reporting the results.

It is crystal clear that this work needs much work in terms of literature review. The number of references is very low. Please try to read and integrate more works.

Reference

Moon, S. J., Hwang, J. S., Kim, J. Y., Shin, A. L., Bae, S. M., & Kim, J. W. (2018). Psychometric properties of the Internet Addiction Test: A systematic review and meta-analysis. Cyberpsychology, Behavior, and Social Networking21(8), 473-484.

Author Response

Article

Portuguese validation of a reduced version of the IAT (Internet Addiction Test) scale - Youth version

- Revision 1 -

Dear Reviewer,

Firstly, we would like to thank you for taking the time and effort necessary to provide insightful guidance, which has contributed to improving this new version of the paper. We carefully considered your comments. Herein, we explain how we revised the manuscript based on those comments and recommendations.

Comment 1: The title could be revised to: ‘’Portuguese validation of a reduced version of the IAT (Internet Addiction Test) scale - Youth version’’

The title has been changed.

Comment 2: Introduction:

- Overall, Intro is too short and it should include more critical information. For instance, there is no explanation on whether the IAT has been examined in other cultures and countries; what were the results? This is very important that authors provide a literature review of the IAT in the Introduction.

- The list of the most prevalent characteristics (Patrão et al., 2023) of young people who present online and smartphone addiction is not adequate in this form. It might be better in a table, but it would be more useful to go in-depth into the psychometric results: e.g. referring to articles such as Moon et al., 2018.

-What did the original study yield on the psychometrics of the measure? Did consequent studies replicate these results? Or was there any need for modification/item removal? All these are needed to be included in the Introduction.

-Have previous studies shown cross-cultural differences in IAT construct?

- Please also provide some hypotheses on what you expect about the factor structure and youth version of IAT and statistics that you have used to examine the validity of the IAT.

The prevalent characteristics of young people with online and smartphone addiction have been tabulated (Table 1)

A table has been added with several studies carried out in various countries. One of the studies compares three countries (China, Malaysia and Spain) with three populations (Table 3)

Comment 3: - The item selection based on the highest factor loadings is not necessarily the best solution, as the youngest in that sample was 15 years old, and here it is 12. The reliability of self-reporting at age 12 is questionable, so the validity of the IAT should be much better supported, because the 7-item factor structure is not enough to confirm the validity of the IAT. It would have been worthwhile even to compare the measurement invariance of the 12-14 years old group with that of the older ones. I think this is more important than gender differences.    

The analysis of invariance about age was not carried out because it was considered more important to consider gender for the analysis of invariance. In other studies with a young population (as in the case of the MBI, the invariance analysis was carried out according to the participant's gender).

Comment 4: -The demographic variables that are compared across groups are interesting. But please first provide information about these variables in the introduction and explain how they are related to the IAT construct. It would be strange for a reader to see these analyses while there are no explanations in the Introduction.

Added to the introduction.

Comment 5: - The prevalence of Internet addiction in the sample is not evaluated, as the diagnostic reliability of the 7-item IAT scale is unknown. In the further results section, it would be more appropriate to write about the validity of this measure among early adolescents (12-14 year olds). It could have been supported by qualitative research (e.g. cognitive interview).

One of the authors of this research is a clinical psychologist. From her practice with young adolescents, she concluded that these were the most important items for assessing internet addiction.

Comment 6: Since normality is violated for several items, it is also important to describe the method of factor analysis (eg. MLR).

The normality of the items is not grossly violated since the absolute values of asymmetry and kurtosis are below 3 and 7, respectively, according to Kline (2011).

Comment 7: Discussion:

- Discussion lacks depth and is too weak. The findings should be interpreted in the context of the results from previous studies. In the current format, the Discussion is mostly a repetition of the results section. What results did previous studies on the psychometric of the IAT yield? What about convergent validity? A literature review on the IAT is needed in the Introduction which could be used to discuss the results from the current study in the Discussion part.

- Please discuss your findings in the context of previous findings while also trying to avoid only reporting the results.

The discussion has been improved.

Comment 8: It is crystal clear that this work needs much work in terms of literature review. The number of references is very low. Please try to read and integrate more works.

New studies were included in the literature review.

Comment 9: Reference

Moon, S. J., Hwang, J. S., Kim, J. Y., Shin, A. L., Bae, S. M., & Kim, J. W. (2018). Psychometric properties of the Internet Addiction Test: A systematic review and meta-analysis. Cyberpsychology, Behavior, and Social Networking, 21(8), 473-484.

The suggested reference has been added to this study.

In closing, we would like to thank you again for your comments. We hope that we have dealt with your suggestions satisfactorily and made all the adjustments requested, both in form and substance.

Yours sincerely,

On behalf of my co-authors,

References added to the manuscript:

(Ali et al., 2021) Ali, Amira Mohammed, Amira Mohammed Ali, Amin Omar Hendawy, Abdulaziz Mofdy Almarwani , Naif Alzahrani, Nashwa Ibrahim, Abdulmajeed A Alkhamees, and Hiroshi Kunugi. 2021. The Six-Item Version of the Internet Addiction Test: Its Development, Psychometric Properties, and Measurement Invariance among Women with Eating Disorders and Healthy School and University Students. International journal of environmental research and public health, 18(23), 12341. https://doi.org/10.3390/ijerph182312341

(Barke et al., 2012) Barke, Antonia; Nyenhuis, Nele; Kröner-Herwig, and Birgit. 2012. The German Version of the Internet Addiction Test: A Validation Study. Cyberpsychology, Behavior, and Social Networking, 15(10), 534–542. doi:10.1089/cyber.2011.0616

(Cernja et al., 2019) Cernja, Iva, Lucija Vejmelka, and Miroslav Rajter. 2019. Internet addiction test: Croatian preliminary study. BMC Psychiatry, 19, 388

(Diotaiuti et al., 2022) Diotaiuti, Pierluigi, Stefania Mancone, Stefano Corrado, Alfredo De Risio, Elisa Cavicchiolo, Laura Girelli, and Andrea Chirico. 2022. Internet addiction in young adults: The role of impulsivity and codependency. Frontiers in psychiatry, 13, 893861. https://doi.org/10.3389/fpsyt.2022.893861

(Durkee et al., 2012) Durkee, Tony, Michael Kaess, Vladimir Carli, Peter Parzer, Camilla Wasserman, Birgitta Floderus, Alan Apter, Judit Balazs, Shira Barzilay, Julio Bobes, Romuald Brunner, Paul Corcoran, Doina Cosman, Padraig Cotter, Romain Despalins, Nadja Graber, Francis Guillemin, Christian Haring, Jean-Pierre Kahn, Laura Mandelli, Dragan Marusic, Gergely Mészáros, George J Musa, Vita Postuvan, Franz Resch, Pilar A Saiz, Merike Sisask, Airi Varnik, Marco Sarchiapone, Christina W Hoven, and Danuta Wasserman. 2012. Prevalence of pathological internet use among adolescents in Europe: demographic and social factors. Addiction (Abingdon, England), 107(12), 2210–2222. https://doi.org/10.1111/j.1360-0443.2012.03946.x

(Fernandez-Villa et al., 2015) Fernandez-Villa, Tânia, Antonio J. Molina, Miguel García-Martín, Javier Llorca, Miguel Delgado-Rodríguez, and Vicente Martín. 2015. Validation and psychometric analysis of the Internet Addiction Test in Spanish among college students. BMC Public Health, 15, 953.

(Kaya et al., 2016) Kaya, Fatih, Erhan Delen, and Kimberly S Young. 2016. Psychometric properties of the Internet Addiction Test in Turkish. J. Behav. Addict., 5, 130–134.

(Khazaal et al., 2008) Khazaal, Yasser, Joël Billieux, Gabriel Thorens, Riaz Khan, Youssr Louati, Elisa Scarlatti, Florence Theintz, Jerome Lederrey, Martial Van Der Linden, and Daniele Zullino. 2008. French validation of the internet addiction test. Cyberpsychol. Behav. Impact Internet Multimed. Virtual Real. Behav. Soc., 11, 703–706.

(Lam et al., 2009) Lam, Lawrence T., Zi-wen Peng, Jin-cheng Mai, and Jin Jing. 2009. Factors associated with Internet addiction among adolescents. Cyberpsychol. Behav. Impact Internet Multimed. Virtual Real. Behav. Soc., 12, 551–555.

(Moon et al., 2018) Moon, Sun Jae, Jin Seub Hwang, Jae Yup Kim, Ah Lahm Shin, Seung Min Bae, and Jung Won Kim. 2018. Psychometric Properties of the Internet Addiction Test: A Systematic Review and Meta-Analysis. Cyberpsychol. Behav. Soc. Netw., 21, 473–484

(Neelapaijit et al., 2018) Neelapaijit, Adam, Manee Pinyopornpanish, Sutapat Simcharoen, Pimolpun Kuntawong, Nahathai Wongpakaran, and Tinakon Wongpakaran. 2018. Psychometric properties of a Thai version internet addiction test. BMC Res. Notes, 11, 69

(Odaci and Cikrikci, 2017) Odaci, H.atice, and Ozkan Cikrikci. 2017. An exploration of the associations among internet use, Depression, anxiety and stress among youths. Mediterranean Journal of Clinical Psychology, 5 , 1-13. https://doi.org/10.6092/2282-1619/201.5.1635

(Rial et al., 2015) Rial, Antonio, Patricia Gómez, Isorna Folgar Manuel, Manuel Araujo Gallego, and J. Varela-Mallou. 2015. EUPI-a: Escala de Uso Problemático de Internet en adolescentes. Desarrollo y validación psicométrica. Adicciones, 27, 47–63.

(Rial et al., 2018) Rial, Antonio, Sandra Golpe, Manuel Isorna, Teresa Braña, and Patricia Gómez. 2018. Minors and problematic internet use: Evidence for better prevention. Computers in Human Behavior, 87, 140–145. https://doi.org/10.1016/j.chb.2018.05.030

(Rosenthal et al., 2018) Rosenthal, Samantha R., Yoojin Cha, and Melissa A. Clark. 2018. The Internet Addiction Test in a Young Adult U.S. Population. Health & Wellness Department Faculty Publications and Research. 5. https://scholarsarchive.jwu.edu/health_fac/5

(Samaha et al., 2018) Samaha, Ali A., Mirna Fawaz, Najwa El Yahfoufi, Maya Gebbawi, Hassan Abdallah , Safaa A Baydoun , Ali Ghaddar , and Ali H Eid. 2018. Assessing the Psychometric Properties of the Internet Addiction Test (IAT) Among Lebanese College Students. Front. Public Health 6:365. doi: 10.3389/fpubh.2018.00365

(Siomos et al., 2008) Siomos, Konstantinos E., Evaggelia D Dafouli, Dimitrios A Braimiotis, Odysseas D Mouzas, and Nikiforos V Angelopoulos. 2008. Internet addiction among Greek adolescent students. Cyberpsychol. Behav. Impact Internet Multimed. Virtual Real. Behav. Soc., 11, 653–657.

(Siste et al., 2021) Siste, Kristiana, Christiany Suwartono, Martina Wiwie Nasrun, Saptawati Bardosono, Rini Sekartini, Jacub Pandelaki, Riza Sarasvita, Belinda Julivia Murtani, Reza Damayanti, and Tjhin Wiguna. 2021. Validation study of the Indonesian internet addiction test among adolescents. PLoS ONE, 16, e0245833.

(Strong et al., 2018) Strong, Carol, Chih-Ting Lee, Lo-Hsin Chao, Chung-Ying Lin, and Meng-Che Tsai. 2018. Adolescent Internet Use, Social Integration, and Depressive Symptoms:Analysis from a Longitudinal Cohort Survey. Journal of Developmental& Behavioral Pediatrics, 39, 318-324. pmid:29461298

(Tateno et al., 2018) Tateno, Masaru, Alan R Teo, Masaki Shiraishi, Masaya Tayama, Chiaki Kawanishi, and Takahiro A Kato. 2018. Prevalence rate of Internet addiction among Japanese college students: Two cross-sectional studies and reconsideration of cut-off points of Young’s Internet Addiction Test in Japan. Psychiatry Clin. Neurosci., 72, 723–730.

(Valenti et al., 2023) Valenti, Giusy D., Giuseppe Craparo, and Palmira Faraci. 2023. The Development of a Short Version of the Internet Addiction Test: The IAT-7. Int J Ment Health Addiction (2023). https://doi.org/10.1007/s11469-023-01153-4

Reviewer 2 Report

Comments and Suggestions for Authors

Modifications or suggestions for improvement

In the literature review I suggest addressing the controversy surrounding the conceptualisation of internet addiction (not captured in the DSM5 or ICD11) in a little more detail and addressing the concept of Problematic Internet Use, as well as further reflection on the health implications of this, specifically incorporating some references: Durkee et al. (2012), Strong et al. (2018), Odaci & Cikricki (2017) or Rial, Golpe, Isorna, Braña & Gómez (2018).

On the other hand, I also suggest that the authors carry out an in-depth review of the scales for assessing problematic internet use on a European level, for example the EUPI-a by Rial et al. (2015) has not been added.

At the methodological level, there is a lack of relevant data on the participants, how many schools, how and by whom the questionnaires were collected, whether anonymity and confidentiality were guaranteed, whether parental permission was sought, whether permission was obtained from the bioethics committee, how many subjects did not respond, how many questionnaires were rejected. Without these requirements this article cannot be published.

Another methodological issue I would suggest, first of all, to emphasise that this is an exploratory study and that the data should not be interpreted in terms of prevalence (I would not speak of "prevalence"), especially when a sample of only 3021 subjects is used, given the important limitations from the point of view of external validity.

At the level of data analysis, I suggest "the possibility" of incorporating some type of analysis (e.g. Logistic Regression) that allows us to quantify the prognostic or discriminatory capacity of the pattern of "problematic Internet use" with respect to some variable such as cyberbullying, school failure, family problems, sedentary lifestyle, etc.

I do not note that the authors establish a cut-off point for the "Internet addiction" or Problematic Internet Use score. The authors should provide the sensitivity of the scale and its specificity. In other words, the screening instrument should be able to detect true positives of cases and to reject true negatives, both of which are very important for the validation of a scale. In a complementary way, they should perform a Receiver Operating Characteristic (ROC) curve analysis, to observe the area under the curve.

In the Discussion I consider that the idea of the need to carry out early detection of Problematic Internet Use as a prevention and screening strategy should be incorporated/reinforced.

The Conclusions are "excessively" concise. I suggest a greater effort to develop this section.

Author Response

Article

Portuguese validation of a reduced version of the IAT (Internet Addiction Test) scale - Youth version

- Revision 1 -

Dear Reviewer,

Firstly, we would like to thank you for taking the time and effort necessary to provide insightful guidance, which has contributed to improving this new version of the paper. We carefully considered your comments. Herein, we explain how we revised the manuscript based on those comments and recommendations.

Comment 1: In the literature review I suggest addressing the controversy surrounding the conceptualisation of internet addiction (not captured in the DSM5 or ICD11) in a little more detail and addressing the concept of Problematic Internet Use, as well as further reflection on the health implications of this, specifically incorporating some references: Durkee et al. (2012), Strong et al. (2018), Odaci & Cikricki (2017) or Rial, Golpe, Isorna, Braña & Gómez (2018).

All these references have been added to the study.

Comment 2: On the other hand, I also suggest that the authors carry out an in-depth review of the scales for assessing problematic internet use on a European level, for example the EUPI-a by Rial et al. (2015) has not been added.

This scale has been added to the table of instruments.

Comment 3: At the methodological level, there is a lack of relevant data on the participants, how many schools, how and by whom the questionnaires were collected, whether anonymity and confidentiality were guaranteed, whether parental permission was sought, whether permission was obtained from the bioethics committee, how many subjects did not respond, how many questionnaires were rejected. Without these requirements this article cannot be published.

The number of school groupings has been added to the procedure.

The Ethics Committee's opinion number was added to the procedure.

The procedure states that the confidentiality of the data was guaranteed, and that the questionnaire was posted online on the Google Forms platform.

The procedure states that in the case of minors, the parents signed the informed consent.

Comment 4: Another methodological issue I would suggest, first of all, to emphasise that this is an exploratory study and that the data should not be interpreted in terms of prevalence (I would not speak of "prevalence"), especially when a sample of only 3021 subjects is used, given the important limitations from the point of view of external validity.

Thanks for the suggestion. We've tried to make some changes.

Comment 5: At the level of data analysis, I suggest "the possibility" of incorporating some type of analysis (e.g. Logistic Regression) that allows us to quantify the prognostic or discriminatory capacity of the pattern of "problematic Internet use" with respect to some variable such as cyberbullying, school failure, family problems, sedentary lifestyle, etc.

Thanks for the suggestion. However, the data you are asking for is not part of the questionnaire used in this study. That's why we don't have this data: cyberbullying, school failure, family problems, physical inactivity. It wasn't our aim to evaluate this data. We know that there is discrimination according to level of education, but we didn't consider it relevant to include in the study.

Comment 6: I do not note that the authors establish a cut-off point for the "Internet addiction" or Problematic Internet Use score. The authors should provide the sensitivity of the scale and its specificity. In other words, the screening instrument should be able to detect true positives of cases and to reject true negatives, both of which are very important for the validation of a scale. In a complementary way, they should perform a Receiver Operating Characteristic (ROC) curve analysis, to observe the area under the curve.

The cut-off points were calculated proportionally to the cut-off points of the 20-item IAT scale. As for the normality of the scale, a sentence was added to the results.

Comment 7: In the Discussion I consider that the idea of the need to carry out early detection of Problematic Internet Use as a prevention and screening strategy should be incorporated/reinforced.

It was added to the discussion.

Comment 8: The Conclusions are "excessively" concise. I suggest a greater effort to develop this section.

The conclusions have been improved.

In closing, we would like to thank you again for your comments. We hope that we have dealt with your suggestions satisfactorily and made all the adjustments requested, both in form and substance.

Yours sincerely,

On behalf of my co-authors,

References added to the manuscript:

(Ali et al., 2021) Ali, Amira Mohammed, Amira Mohammed Ali, Amin Omar Hendawy, Abdulaziz Mofdy Almarwani , Naif Alzahrani, Nashwa Ibrahim, Abdulmajeed A Alkhamees, and Hiroshi Kunugi. 2021. The Six-Item Version of the Internet Addiction Test: Its Development, Psychometric Properties, and Measurement Invariance among Women with Eating Disorders and Healthy School and University Students. International journal of environmental research and public health, 18(23), 12341. https://doi.org/10.3390/ijerph182312341

(Barke et al., 2012) Barke, Antonia; Nyenhuis, Nele; Kröner-Herwig, and Birgit. 2012. The German Version of the Internet Addiction Test: A Validation Study. Cyberpsychology, Behavior, and Social Networking, 15(10), 534–542. doi:10.1089/cyber.2011.0616

(Cernja et al., 2019) Cernja, Iva, Lucija Vejmelka, and Miroslav Rajter. 2019. Internet addiction test: Croatian preliminary study. BMC Psychiatry, 19, 388

(Diotaiuti et al., 2022) Diotaiuti, Pierluigi, Stefania Mancone, Stefano Corrado, Alfredo De Risio, Elisa Cavicchiolo, Laura Girelli, and Andrea Chirico. 2022. Internet addiction in young adults: The role of impulsivity and codependency. Frontiers in psychiatry, 13, 893861. https://doi.org/10.3389/fpsyt.2022.893861

(Durkee et al., 2012) Durkee, Tony, Michael Kaess, Vladimir Carli, Peter Parzer, Camilla Wasserman, Birgitta Floderus, Alan Apter, Judit Balazs, Shira Barzilay, Julio Bobes, Romuald Brunner, Paul Corcoran, Doina Cosman, Padraig Cotter, Romain Despalins, Nadja Graber, Francis Guillemin, Christian Haring, Jean-Pierre Kahn, Laura Mandelli, Dragan Marusic, Gergely Mészáros, George J Musa, Vita Postuvan, Franz Resch, Pilar A Saiz, Merike Sisask, Airi Varnik, Marco Sarchiapone, Christina W Hoven, and Danuta Wasserman. 2012. Prevalence of pathological internet use among adolescents in Europe: demographic and social factors. Addiction (Abingdon, England), 107(12), 2210–2222. https://doi.org/10.1111/j.1360-0443.2012.03946.x

(Fernandez-Villa et al., 2015) Fernandez-Villa, Tânia, Antonio J. Molina, Miguel García-Martín, Javier Llorca, Miguel Delgado-Rodríguez, and Vicente Martín. 2015. Validation and psychometric analysis of the Internet Addiction Test in Spanish among college students. BMC Public Health, 15, 953.

(Kaya et al., 2016) Kaya, Fatih, Erhan Delen, and Kimberly S Young. 2016. Psychometric properties of the Internet Addiction Test in Turkish. J. Behav. Addict., 5, 130–134.

(Khazaal et al., 2008) Khazaal, Yasser, Joël Billieux, Gabriel Thorens, Riaz Khan, Youssr Louati, Elisa Scarlatti, Florence Theintz, Jerome Lederrey, Martial Van Der Linden, and Daniele Zullino. 2008. French validation of the internet addiction test. Cyberpsychol. Behav. Impact Internet Multimed. Virtual Real. Behav. Soc., 11, 703–706.

(Lam et al., 2009) Lam, Lawrence T., Zi-wen Peng, Jin-cheng Mai, and Jin Jing. 2009. Factors associated with Internet addiction among adolescents. Cyberpsychol. Behav. Impact Internet Multimed. Virtual Real. Behav. Soc., 12, 551–555.

(Moon et al., 2018) Moon, Sun Jae, Jin Seub Hwang, Jae Yup Kim, Ah Lahm Shin, Seung Min Bae, and Jung Won Kim. 2018. Psychometric Properties of the Internet Addiction Test: A Systematic Review and Meta-Analysis. Cyberpsychol. Behav. Soc. Netw., 21, 473–484

(Neelapaijit et al., 2018) Neelapaijit, Adam, Manee Pinyopornpanish, Sutapat Simcharoen, Pimolpun Kuntawong, Nahathai Wongpakaran, and Tinakon Wongpakaran. 2018. Psychometric properties of a Thai version internet addiction test. BMC Res. Notes, 11, 69

(Odaci and Cikrikci, 2017) Odaci, H.atice, and Ozkan Cikrikci. 2017. An exploration of the associations among internet use, Depression, anxiety and stress among youths. Mediterranean Journal of Clinical Psychology, 5 , 1-13. https://doi.org/10.6092/2282-1619/201.5.1635

(Rial et al., 2015) Rial, Antonio, Patricia Gómez, Isorna Folgar Manuel, Manuel Araujo Gallego, and J. Varela-Mallou. 2015. EUPI-a: Escala de Uso Problemático de Internet en adolescentes. Desarrollo y validación psicométrica. Adicciones, 27, 47–63.

(Rial et al., 2018) Rial, Antonio, Sandra Golpe, Manuel Isorna, Teresa Braña, and Patricia Gómez. 2018. Minors and problematic internet use: Evidence for better prevention. Computers in Human Behavior, 87, 140–145. https://doi.org/10.1016/j.chb.2018.05.030

(Rosenthal et al., 2018) Rosenthal, Samantha R., Yoojin Cha, and Melissa A. Clark. 2018. The Internet Addiction Test in a Young Adult U.S. Population. Health & Wellness Department Faculty Publications and Research. 5. https://scholarsarchive.jwu.edu/health_fac/5

(Samaha et al., 2018) Samaha, Ali A., Mirna Fawaz, Najwa El Yahfoufi, Maya Gebbawi, Hassan Abdallah , Safaa A Baydoun , Ali Ghaddar , and Ali H Eid. 2018. Assessing the Psychometric Properties of the Internet Addiction Test (IAT) Among Lebanese College Students. Front. Public Health 6:365. doi: 10.3389/fpubh.2018.00365

(Siomos et al., 2008) Siomos, Konstantinos E., Evaggelia D Dafouli, Dimitrios A Braimiotis, Odysseas D Mouzas, and Nikiforos V Angelopoulos. 2008. Internet addiction among Greek adolescent students. Cyberpsychol. Behav. Impact Internet Multimed. Virtual Real. Behav. Soc., 11, 653–657.

(Siste et al., 2021) Siste, Kristiana, Christiany Suwartono, Martina Wiwie Nasrun, Saptawati Bardosono, Rini Sekartini, Jacub Pandelaki, Riza Sarasvita, Belinda Julivia Murtani, Reza Damayanti, and Tjhin Wiguna. 2021. Validation study of the Indonesian internet addiction test among adolescents. PLoS ONE, 16, e0245833.

(Strong et al., 2018) Strong, Carol, Chih-Ting Lee, Lo-Hsin Chao, Chung-Ying Lin, and Meng-Che Tsai. 2018. Adolescent Internet Use, Social Integration, and Depressive Symptoms: Analysis from a Longitudinal Cohort Survey. Journal of Developmental& Behavioral Pediatrics, 39, 318-324. pmid:29461298

(Tateno et al., 2018) Tateno, Masaru, Alan R Teo, Masaki Shiraishi, Masaya Tayama, Chiaki Kawanishi, and Takahiro A Kato. 2018. Prevalence rate of Internet addiction among Japanese college students: Two cross-sectional studies and reconsideration of cut-off points of Young’s Internet Addiction Test in Japan. Psychiatry Clin. Neurosci., 72, 723–730.

(Valenti et al., 2023) Valenti, Giusy D., Giuseppe Craparo, and Palmira Faraci. 2023. The Development of a Short Version of the Internet Addiction Test: The IAT-7. Int J Ment Health Addiction (2023). https://doi.org/10.1007/s11469-023-01153-4

Round 2

Reviewer 1 Report

Comments and Suggestions for Authors

Thank you for the corrections. I consider the amended manuscript acceptable. However, I have three comments:

1. I do not fully understand why two samples had to be used for the CFA analysis. It would be worthwhile to justify this and support it with literature. The sample is usually divided into two parts in order to perform the EFA on one and the CFA on the other.

 2. I don't think the following sentence is good, I couldn't find the quoted content in the 2010 edition. "Concerning item sensitivity, it was found that none of the items has a median close to one of the extremes, all items have responses in all points, and their absolute values of skewness and kurtosis are below 3 and 8, respectively (Table 6), which indicates that they do not grossly violate normality [49]. As far as the scale is concerned, it does not grossly violate normality since its absolute skewness and kurtosis values are below 3 and 8, respectively [49]."

The latest version (Kline, 2023, pp. 60) has the following: "Just as there are no golden rules for detecting outliers, there are also no universal absolute values for skewness or kurtosis statistics that indicate severe nonnormality. Finney and DiStefano (2013) noted that absolute univariate skewness and kurtosis values greater than, respectively, 2.0 and 7.0, have been described as indicating severe nonnormality in some computer simulation studies, but exceptions are easy to find. For example, Lei and Lomax (2005) treated absolute skewness and kurtosis values > 2.30 as indicating severe nonnormality. The point is that there is no magic demarcation between trivial and appreciable nonnormality that will fit all models and data sets, but the assumption of normality becomes increasingly less plausible as there is more and more skewness or kurtosis."

Since the skewness and kurtosis in the study are certainly not a problem (none exceeds 2), the problem of nonnormality in factor analysis should not be feared; however, the above statement should be clarified.

3. I feel critical about whether the questionnaire developed for adults in the 12–14 age group is accessible to adolescents. There are items that most probably need rewording: e.g., Does your job performance or productivity suffer because of the Internet?   

Author Response

Article

Portuguese validation of a reduced version of the IAT (Internet Addiction Test) scale - Youth version

- Revision 2 -

Dear Reviewer,

Firstly, we would like to thank you for taking the time and effort necessary to provide insightful guidance, which has contributed to improving this new version of the paper. We carefully considered your comments. Herein, we explain how we revised the manuscript based on those comments and recommendations.

Comment 1: I do not fully understand why two samples had to be used for the CFA analysis. It would be worthwhile to justify this and support it with literature. The sample is usually divided into two parts in order to perform the EFA on one and the CFA on the other.

Page 5 (lines 163-167) describes why we divided the sample into three parts (20%, 40% and 40%) and is referenced by Marôco (2021).

Comment 2: I don't think the following sentence is good, I couldn't find the quoted content in the 2010 edition. "Concerning item sensitivity, it was found that none of the items has a median close to one of the extremes, all items have responses in all points, and their absolute values of skewness and kurtosis are below 3 and 8, respectively (Table 6), which indicates that they do not grossly violate normality [49]. As far as the scale is concerned, it does not grossly violate normality since its absolute skewness and kurtosis values are below 3 and 8, respectively [49]."

The latest version (Kline, 2023, pp. 60) has the following: "Just as there are no golden rules for detecting outliers, there are also no universal absolute values for skewness or kurtosis statistics that indicate severe nonnormality. Finney and DiStefano (2013) noted that absolute univariate skewness and kurtosis values greater than, respectively, 2.0 and 7.0, have been described as indicating severe nonnormality in some computer simulation studies, but exceptions are easy to find. For example, Lei and Lomax (2005) treated absolute skewness and kurtosis values > 2.30 as indicating severe nonnormality. The point is that there is no magic demarcation between trivial and appreciable nonnormality that will fit all models and data sets, but the assumption of normality becomes increasingly less plausible as there is more and more skewness or kurtosis."

Since the skewness and kurtosis in the study are certainly not a problem (none exceeds 2), the problem of nonnormality in factor analysis should not be feared; however, the above statement should be clarified.

Thank you for your suggestion. I looked up the article referred to by Kline (2023) and changed the absolute values of asymmetry and kurtosis and the respective reference.

Comment 3: - I feel critical about whether the questionnaire developed for adults in the 12–14 age group is accessible to adolescents. There are items that most probably need rewording: e.g., Does your job performance or productivity suffer because of the Internet? 

Thank you for noticing such a big mistake. We inexplicably mistranslated the scale from Portuguese to English. I have corrected this item.

In closing, we would like to thank you again for your comments. We hope that we have dealt with your suggestions satisfactorily and made all the adjustments requested, both in form and substance.

Yours sincerely,

On behalf of my co-authors,

Reference removed from the manuscript:

Kline, R. B. Principles and Practice of Structural Equation Modeling (3rd ed.). 2011. New York: Guilford Press.

References added to the manuscript:

Finney, S. J., and DiStefano, C. Nonnormal and categorical data in structural equation modeling. In G. R. Hancock & R. O. Mueller (Eds.), Structural equation modeling: A second course (pp. 439–492). 2013. IAP Information Age Publishing.

Reviewer 2 Report

Comments and Suggestions for Authors

I believe that with the modifications made, the article has gained a lot of quality and meets the requirements for publication. I congratulate the authors for their efforts.

Author Response

Article

Portuguese validation of a reduced version of the IAT (Internet Addiction Test) scale - Youth version

- Revision 2 -

Dear Reviewer,

Firstly, we would like to thank you for taking the time and effort necessary to provide insightful guidance, which has contributed to improving this new version of the paper. We carefully considered your comments. Herein, we explain how we revised the manuscript based on those comments and recommendations.

Comment 1: I believe that with the modifications made, the article has gained a lot of quality and meets the requirements for publication. I congratulate the authors for their efforts.

Thanks for the excellent reply. We tried hard to improve the manuscript.

In closing, we would like to thank you again for your comments. We hope that we have dealt with your suggestions satisfactorily and made all the adjustments requested, both in form and substance.

Yours sincerely,

On behalf of my co-authors,

Reference removed from the manuscript:

Kline, R. B. Principles and Practice of Structural Equation Modeling (3rd ed.). 2011. New York: Guilford Press.

References added to the manuscript:

Finney, S. J., and DiStefano, C. Nonnormal and categorical data in structural equation modeling. In G. R. Hancock & R. O. Mueller (Eds.), Structural equation modeling: A second course (pp. 439–492). 2013. IAP Information Age Publishing.
